# Bacteria, Fungi and Archaea Domains in Rhizospheric Soil and Their Effects in Enhancing Agricultural Productivity

**DOI:** 10.3390/ijerph16203873

**Published:** 2019-10-12

**Authors:** Kehinde Abraham Odelade, Olubukola Oluranti Babalola

**Affiliations:** Food Security and Safety Niche Area, Faculty of Natural and Agricultural Science, North-West University, Private Bag X2046, Mmabatho 2735, South Africa; kennybramm@gmail.com

**Keywords:** enhance plant growth, host plant growth, rhizosphere microbiome, improve crop productions and suitable eco-friendly options

## Abstract

The persistent and undiscriminating application of chemicals as means to improve crop growth, development and yields for several years has become problematic to agricultural sustainability because of the adverse effects these chemicals have on the produce, consumers and beneficial microbes in the ecosystem. Therefore, for agricultural productivity to be sustained there are needs for better and suitable preferences which would be friendly to the ecosystem. The use of microbial metabolites has become an attractive and more feasible preference because they are versatile, degradable and ecofriendly, unlike chemicals. In order to achieve this aim, it is then imperative to explore microbes that are very close to the root of a plant, especially where they are more concentrated and have efficient activities called the rhizosphere. Extensive varieties of bacteria, archaea, fungi and other microbes are found inhabiting the rhizosphere with various interactions with the plant host. Therefore, this review explores various beneficial microbes such as bacteria, fungi and archaea and their roles in the environment in terms of acquisition of nutrients for plants for the purposes of plant growth and health. It also discusses the effect of root exudate on the rhizosphere microbiome and compares the three domains at molecular levels.

## 1. Introduction

Soil management has become a crucial matter in order to prevent and/or reduce dangers that are being posed to agricultural sustainability in terms of plant development, yield and health. Clearly, management of soil is very vital because of its fragility and renewability which could either affect the ecosystem positively or negatively. Limitations of the climatic conditions and differences in the gradient of the soil are responsible for its unequal distribution across the geographical zones of the earth; it is immovable and cannot be transported from one place to another [1]. Among other various roles soil can perform is its ability to act as an environment that enables plant cultivation possible in order to produce crops that serve as food for man and animal populace consumption. The health of man and animals are solely dependent on the quality of the soil because it acts as the paramount means of producing fibrous crops and food. The qualitative content of the air and water that are being consumed directly depend on the conditions of the soil as it is a significant link to the environment at large. Hence, the quality of the soil is directly proportional to the health of the living things and environment. The rate at which the land is being used and its sustainability can be measured by the health status of the soil contents [2]. Conventionally, the quality of a soil has been focused and linked with the type of agricultural system that gives us maximal yield. Therefore, plant or crop productivity is an important pointer to determining the quality of the soil and also the recommendable agricultural practice. Soil management and contents of the soil such as nutrients, germplasm, water, pathogens and dissolved oxygen can be bio-assayed by the yield of the crop. 

The population of microbes found colonizing the plant environment can be as much greater than the cell concentrations of the plant itself. A quantum number of research works have shown the significant effects microbes affiliated with plants could have on growth, development, effective germination of seed, seedling strength, diseases and nutrition. The entire microbial genome inhabiting a plant environment is described as the microbiome of a plant or is called plants microbial genome. In the context of this review, plants are described as versatile organisms, but that depends on the inhabiting microbes for definite roles and characters. Consequently, plants via photosynthesis manufacture their own food and deposit the fixed carbon into spermospheric, phyllospheric, rhizospheric and mycorrhizospheric surroundings directly [3,4]. The availability of the carbon at a particular area in the surroundings plays a great role in determining the microbial community, composition, structure and activities. Up till now, there have been various in depth studies on the interactions between plants and microbes, especially in the areas of rhizobacteria symbiotic and leaf pathogenic associations. Nevertheless, knowledge about the impacts of a majority of these microbes in the plant-microbes association, especially in terms of the growth, health and disease of plants, is limited. Therefore, decoding the diversity of the microbiome present in the plant is very paramount for the purpose of identifying microbes that could be of great exploitation in the growth and health improvement of plants.

However, system productivity cannot be used as a perfect measure for the crop yield. Rather, both the system productivity as well as its function are better representations for measuring soil quality. The soil properties with complex interactions that enhance the quality of soil and its ability to function are the following: Biological, chemical and physical properties. All these properties are the indicators of the quality of soil that could be used to measure the soil capacity and the function it performs [5]. The indicators for soil quality could be adapted to measure changes triggered by the management of soil and crop practices. Generally, in the world today there have been so many works to analyze soil quality, and thereby a number of chemical and physical indicators have been identified [6]. Interestingly, biological properties of soil have been proven to be the determining factor. Directly or indirectly, a majority of the indicators are functions of the activities of different microbial domains present in the rhizosphere.

## 2. The Rhizosphere

The term rhizosphere originated from the words rhiza and sphere, which mean root and environment of influence, respectively. In 1904, the term was coined by an agronomist whose area of specialization was plant physiology, a national of Germany called Lorenz Hiltner. Rhizosphere describes an interface between the plant and its root [7]. Hiltner was the first scientist to describe the rhizosphere as the area with high proximity (1–2 mm thick) to the root of a plant colonized by a unique quantum number of microbiome with economic importance, and which is influenced through the plant root organic compounds released during chemical reactions. The main active part of the soil which involves a biogeochemical mechanism of reaction for the purpose of influencing landscape host and global scale practices is referred to as the rhizosphere. For the wise maintenance of planet health and enrichment of the organisms (microbes, plants, animals and humans) inhabiting it, a better knowledge of understanding these processes is very paramount [8]. In the next fifty years, there will be a great demand in meeting the project of doubling food globally, thus intensive efforts are required to harness the plant root system for potential yields of primary food crop increments [9]. In order to pragmatically face the present challenges of change in universal climate and increase in population, which shall soon mandatorily require more food, feedstuff and fiber agricultural production; these concerted efforts are of paramount importance. Some developing countries are already encountering this condition especially where there is less optimal and habitual non-fertile land [10]. A better understanding of rhizo-microbial metabolisms, including its networking, will be a great frontier in science to tackle these global challenges of change in climate and increase in population. In the subsequent decade it will be so vital that it will require various interdisciplinary proficient workforces. Presently, since rhizosphere metabolic activities directly and indirectly influence plant growth, development and yield, more significant and reasonable considerations have been shifted to it. Through the roots of plant, rhizosphere has a great influence on plant productivity by assisting in the metabolisms of disintegration and cycling of nutrients of soil organic material [11]. The rate at which the soil organic material primarily decomposes depends largely on the presence of living root rhizosphere. The food web activities present in the soil are dependent solely on the following three principal carbon sources: Soil carbon-based (organic) materials, exudates from plant roots and litter or deposits. There is a difference in the way by which these three carbon sources are accessible and available to the soil microbes and hence this dictates within the food webs the increase in the flow of carbon and the diversity of microbes. Soluble complex exudation, uptake of water, plant roots’ and microbes’ nutrient organization, decomposition of soil organic materials via rhizosphere mediation and consequent carbondioxide released via respiration are the processes carried out by the rhizosphere and these involve both the plant roots and inhabiting microbes. Contextually, the major entryways for both water and nutrients are the metabolisms within the roots and native microbes. In the rhizosphere, complex compounds like amino acids, carbohydrates, organic acids and other complexes are released via the root and consequently are used up by inhabiting microbes. In addition, each species of plant discharges its own special signals of complex into the rhizosphere soil. The plant community diversity above the ground is directly proportional to the community diversity below the ground. 

Additionally, it is interesting to know that the number of genes of microbiome present in the rhizosphere is far more than the genes of the plant. Moreover, microbiome found at the rhizosphere are very diverse in structure and composition with great influence on the host plant (in different ways), this may be as a result of the adjacent proximity to the plant root and their constant association. Their association can be widely regarded as a positive effect by improving the growth and health of plants and a negative effect by being harmful (pathogenic) to the plant yield [12]. Hence, understanding the compositional, structural and ecological dynamics as well as the activities of this rhizosphere microbiome is very imperative for the purpose of exploiting them as beneficial tools for agricultural sustainable development goals. 

Furthermore, this zone of the soil (rhizosphere) accommodates a large volume of microbial activities. However, over time now, the definition of rhizosphere has been considered to encompass the quantity of soil that is under the influence of the plant root and root tissue parts. It also encompasses the surrounding soil of the plant root, which the growth and activity of the root have changed biologically, chemically and physically [12]. Most recently, relative proximity has been a yardstick to subdividing the rhizosphere soil into three zones which thus have significant influence on the root; they are:

i. Endorhizospheric zone: This comprises mainly the tissue of the root and also includes the endodermic and cortical layers.

ii. Rhizoplane zone: This is the surface of the root that accommodates the microbes and soil matters. It comprises of mucilaginous and epidermic layers and also include the cortex.

iii. Ectorhizospheric zone: This is comprised of soil closely adhering to the plant root.

In addition to these three main regions, in some cases, definite layers may be well-defined, for instance, the mycorrhizospheric zone, which is the plant root that forms relationship with fungi [13], whereas in other plants, rhizosheath is established; this is the toughly adhering condensed layer. It is made up of root hairs, microbes, soil particles and mucoid materials [14]. The plant root is also a portion of the rhizosphere because endophytic microbes are colonizers of the internal tissues of the root [15]. The quantity of soil that is quite far from the rhizosphere which is not under the influence of the plant root is referred to as bulk soil [16]. During the activity of the rhizosphere, there is transformation of the roots that are dead into soil particles, although this is unlike the bulk soil. Hence, rhizosphere can be said to be an exclusive zone different from the bulk soil.

## 3. The Effects of Rhizosphere 

There is an interaction between the developing plant and a range of microbes inhabiting the soil surrounding the plant during the germination period of the seed and sprouting of seedling. During the seed’s germination into seedling, organic matters are released as the roots grow via the soil and this instigates the activities of the microbes’ diversity in the plant root region and soil surrounding the plant with a little millimeter of thickness. This term is known as the effect of rhizosphere [17]. Hence, the effect of rhizosphere could be regarded as a means of creating an interactive and dynamic environment for microbial development. The calculation is done in terms of rhizosphere and bulk soil ratio, i.e., R:S, and is calculated by dividing the entire population of microbes present in the root rhizosphere (R) by the corresponding quantity of the bulk or surrounding soil (S) [18]. The reflection of the effect of the rhizosphere is noticeable in the structural difference of the microbial diversity present in either cultivated or uncultivated farmland. Rhizosphere linked factors which includes; the characteristics of soil, variety of crop and developmental stages of plant growth are incumbent factors that determine variations in the communities of archaea, bacteria and fungi inhabiting a soil environment [19,20,21]. While bacteria with a greater rhizosphere effect showing R:S values which range from 10 to 100 or greater have been reported, new studies that utilize independent cultivation of microbes present in the soil samples have shown significant rhizosphere effects of fungi in the soil environment [22,23].

The content and qualitative richness of the exudates is a function of the diversity and structure that microbiome would possess. This also determines the kind and degree of interaction of the bacterial, fungal and archaeal species in the plant rhizosphere [24]. Understanding the interaction among the living entities in the soil ecological niche that is the microbiome and the root of the plant, the consumers or predators and macro and meso faunal in the food webs is important for knowing the metabolic processes of the rhizosphere. Due to adaptability, a large number of soil microbiome possess the ability that enables them to associate more efficiently with plant roots and they are able to withstand environmental stresses of rhizospheric plants [25]. Availability of nutrients is directly proportional to the level of microbial association with the rhizosphere, also space and accessibility to water enables how they cordially associate with the root of the plant. This is a continuous practice in the evolutionary trend of microbiome where nutrients are not well sufficient for the competing microbes and this is majorly common in natural ecosystems [26]. The association between the rhizosphere and the microbes is usually instigated via chemical signals from the rhizodeposit released from plant metabolisms. The different parts of a rhizodeposit are root exudates, mucilage, root cap cells and root tissues while the latter is made up of sloughed root hairs and epidermal cells as represented in Figure 1.

## 4. Root Exudation Mechanism

The ubiquitous phenomenon of discharging organic matter from the roots of an active plant into the neighboring soil is referred to as root exudation [26]. Two possible mechanisms are adopted for the discharge of organic compounds by the roots and the exudation rates of the root sensu stricto differ broadly amidst the species and conservational factors [27]. Cellular membranes are the channels via which exudates from a root are transported and secreted into the rhizosphere soil surrounding the plant roots. Both border cells and pseudo-border cells are separated from the border and they are other sources from which chemical products from a plant are released [28]. Nevertheless, it is imperative to understand that methodologically identifying the chemical compositions of the exudates from roots and its concentrations in the rhizosphere soil is difficult [29]. At the time of and after the release of exudates, microbes act on the organic matters and hence enriching and separating it from the roots in the usual environments is challenging. Data have been obtained from hydroponic cultures that are sterile and these revealed the nature and quantitative levels of exudates from roots, but then extrapolating the results to the natural situations is difficult [30]. Furthermore, from this perspective, quantification of exudation from the root was done by measuring the labeled CO_2_ produced in the rhizosphere soil of the plant labeled 14C. Therefore, 12–40% of the total quantity of carbohydrates has been reportedly estimated to be synthesized and released into the root surrounding soil via photosynthesis [31]. However, not only carbon substances are released, different types of compounds that contain nitrogen such as amino acids, nitrate ions and ammonium are released through the roots of a plant into the rhizosphere soil [32].

The following are the compositions of the exudates from plant roots: Amino acids, water soluble sugars and organic acids, though it is also composed of other amino substances, phenolics, vitamins, sugar phosphate esters, hormones and others as summarized in Table 1 using the arabidopsis plant [33,34]. A passive procedure is involved in the discharge of these small molecular weight complexes alongside the sharp concentration slope which habitually occurs between the millimolar range of the integral root cytoplasmic cells and micromolar range of the outer (topsoil) solution. The permeability of a membrane is the factor that determines passive (inert) or direct diffusion which is via the fatty acid bilayer of the plasma membranous sheath. This is dependent on the root cells physiological status and the compounds polarity which facilitate the lipophilic exudate infiltration [35]. The process of exudation efficiency may hence be facilitated through stress features which affect the integrity of a membrane, for instance, stress of exudations, deficiency of nutrients and extremes of temperature [36]. Both quantity and quality of root exudate compositions are assumed to be affected through different environmental conditions such as type of soil, state of oxygen, pH, the microbes present, temperature of the soil, available nutrients and intensity of light. The impact of these conditions on the exudation of roots may be greater than alterations which are a result of species of plants [37]. In the plants at a young stage, an estimated proportion of up to 50% of carbon is released from plant roots, while it is less in the plants at maturity stage in a farmland [38,39]. In addition, the exudate compositions and structure may be different at various stages of growth of the plant. For example, at the stage when a plant is carrying about six leaves, plant root mucilage and carboxylates are numerous than when the plant is at an earlier stage. Furthermore, nitrogen is another nutrient whose cycling in the root zone is of significance and usually is in the form of amino acids, NH^4+^, NO^3−^, sloughed plant roots, lysates of cell and other debris derived from a root. Rhizodeposition has been estimated to amount to 20% N of the total nitrogen present in the plant at the matured stage [40,41]. Moreover, on the plant nutritional status, exudation of a root is greatly dependent on the availability and concentration of oligoelements, namely, Na^+^, Mg^++^ and K^+^. There is stimulation of the main enzymes which are usually very important in the process of glycolysis when the concentrations of some elements are low; these enzymes are actively involved in the regulation of glycolytic pathways in the plant cells and these include pyruvate kinase and phosphofructokinase [42]. The regulation of all the plant biological processes involves each of the micronutrients with similarly essential components of main enzymes. Obviously, in several regions, most especially in the tropical environments that usually consist of soils that are tremendously deficient of oligoelement nutrients or with few available nutrients, this can be considered as a constraint to the growth and development of a plant in these regions [12]. To respond to deficient Fe and P in an environment, some plant species naturally ooze organic acid anions while phytosiderophores are exuded in the environment deficient of Fe and Zn [43]. The process of exudation is a function of the health of any plant. The qualitative and quantitative effects of exudates on the biodiversity of microbes could be beneficial, deleterious or affect the processes of ecological activities in the root rhizosphere which directly influence the plant activities and metabolisms such as the availability of nutrients, persistence of pathogens and rooting configurations [44]. Conversely, the activities of microbes modify the exudation process and patterns of root rhizosphere. Hence, rhizodeposition may be concluded to structurally and functionally influence the aspects of the communities of microbes in the root rhizosphere.

## 5. Archaea, Bacteria and Fungi Domains and Their Comparisons 

As a result of reasonable, important and obvious differences in the structural formation of ribosomal RNA of archaea, they were separated to form the third domain. The 16S rRNA is usually the sequenced molecule of any RNA, and is always found in all the microbes to perform exactly the same fundamental and essential function, i.e., protein production. Since this function is very fundamental and significant to life metabolisms, survival of microbes carrying mutated 16S rRNA molecules will almost be impossible. This would make the nucleotide structure of such to become generationally stable over a very long period of time. Interestingly, the 16S rRNA molecule is sizeable enough to admit definite information that may be present in a microbe and also accommodates considerable total of time for sequencing it. Carl Woese, a renowned microbiologist, developed a new technique for sequencing after several studies from sequencing microbes genetically in 1977. This technique adapts breaking down the RNA of a microbe into fragments that could be aligned with the fragments of another or other microbes for comparison [45]. Therefore, the similarity between the species of a group of microbes is directly proportional to the arrangement of their rRNA pattern [46]. 

This new rRNA technique alignment has been widely used to compare, categorize and differentiate various microbes. From his sequencing results of various species of microbes, a group of methagenogenic microbes with entirely different arrays of RNA from known prokaryotic or eukaryotic microbes were discovered [45]. It was observed that the similarity among these methanogens were much greater than it was with other microbes sequenced, and as a result a novel domain of archaea was proposed by Woese [45]. Experimentally, the results of Woese revealed that archaea domains were of more similarity to eukaryotic microbes (fungi) than prokaryotic microbes (bacteria), although their structural similarity to prokaryotic microbes is greater [47]. Therefore, it could be concluded that both archaea and eukarya domains share more recent ancestral commonality than eukarya and bacteria domains generally [47]. In addition, the occurrence of nucleus development happened after bacteria and the ancestral commonality split [47]. Even though archaea could be said to be prokaryotic, nonetheless eukaryotes are more closely related to them and hence could not be categorized under any of the two domains (bacteria and eukarya); their comparisons are summarized in Table 2 [48]. 

Among the three domains, archaea has the ability to use the ether-linked lipids present in the cell of their membranes, which is specific and unique to them. The ester linkages that are present in both prokaryotes (bacteria) and eukaryotes (fungi) are not as chemically stable as the ether linkages found in the archaea. This is a crucial factor that may be said to contribute to the survival ability of archaea in the extremophilic environments that usually put serious stress on the membranes of their cells, for instance, extreme salinity and heat. In the ecosystem, methanogens contribute substantially with several bacteria that are microbes that derive their energy from methane oxidation. This is because; often in that particular environment they are the primary basis of methane and mainly serve as the original producers. Furthermore, methanogenic archaea contribute immensely in the carbon cycle, methane is produced from the breaking down of carbon which is one of the main greenhouse gases (GHG) [49]. 

## 6. Relationship among the Three Microbial Domains in the Rhizosphere

In order to have the basic and fundamental knowledge of the origin of life, understanding the relationship among the three domains is of great importance. Mostly, metabolic pathways are common between archaea and bacteria and these are the objectives of a majority of the genes present in the microbes that are found in these two domains, whereas in both archaea and eukarya (fungal organisms), most genes involved are for the expression of genomes of microbes that are found under these domains [50]. Both archaea and bacteria have a lone bilayer lipid which makes the cell structural formation of archaea very similar to the bacterial cell that is gram-positive within prokaryota [51]. This bilayer lipid usually consists of a copious sacculus, i.e., exoskeleton with a different composition of chemical constituents [52]. Based on various sequences of gene or protein of homologs of the prokaryotes in several phylogenetic trees (Figure 2), the homologs of archaea and gram-positive bacteria are more closely related to each other [51]. Moreover, in a number of essential proteins like glutamine synthetase I and Hsp70, conserved indels are common to both archaea and gram-positive bacteria [51,53]. Genetic inter-domain transfer has been revealed by the interpretation of the phylogeny of these genes and this may not show the relationship(s) among the organisms [54,55].

Furthermore, because of the antibiotic selection pressure response of archaea, it was suggested that they have their origin from gram-positive bacteria [57]. This suggestion was opined as a result of the ability of archaea to resist a broad range of antibiotics primarily synthesized by bacteria that are gram-positive [51] and that the genes that differentiate archaea from gram-positive bacteria are the main target of the antibiotics. Ultimately, it was suggested that the selective force responsible for the resistance of the antibiotics produced by the gram-positive was sufficient to impact wide-ranging changes in several of the target genes producing antibiotics. Therefore, at present these strains are the representatives of the common ancestral archaea [57]. These traits of selecting antibiotic and other selecting forces for competition is directly responsible for the evolution of archaea and their adaptation to environments that are extremophilic such as very hot, highly saline or acidic regions. They are able to adapt to these extreme environments in the bid of escaping from organisms producing antibiotics and occupying new niches [58,59]. Moreover, this suggestion is in conjunction with other research that investigated the relationships between the structures of protein and whether bacteria that are gram-positive may comprise of the original branching derivations within prokaryota [60].

## 7. The Effects of Bacteria in Making Nutrients Available for Plants

Microbiome found at the plant rhizosphere greatly influences the availability of nutrients to plants. Common examples are the rhizobacteria that can fix nitrogen, like rhizobia and symbiotic plant–fungi mycorrhiza that enhances the uptake of phosphorus [29,61]. There have been several reports on physical restructuring of soil and stability of soil aggregates [62] and inhibition of phytopathogenic-related soil microbes [63,64]. Other microbes which possess nitrogen-fixing genes apart from *Rhizobium* and *Bradyrhizobium* have been reported by various researchers [65] who analyzed cowpea root rhizosphere in the western Amazon and found a large genetic diversity of symbiotic rhizobacteria (i.e., rhizobia) other than the aforementioned. They used 16S rRNA gene sequencing for their analysis and the results showed the presence of *Bradyrhizobium*, *Rhizobium*, *Burkholderia* and *Achromobacter* species which are capable of causing cowpea nodulation and are good nitrogen fixers. However, despite a lot of work on the mechanism of how rhizobia fix nitrogen, conferring it to other significant agricultural plant species has not been possible. Furthermore, there have been indications that a proper understanding of various cellular mechanisms in the genome of both Rhizobium and mycorrhizal fungi would enable researchers to achieve the long term goal [66]. Another good examples of rhizospheric microbiome that acquire nutrients for the plants are the ones that possess the ability to make iron readily available for the plant by breaking down the insoluble and toxic ferric iron oxide in the soil. Ferric iron oxide in the soil is between neutral to alkaline at high concentrations and toxic at that state to both microbes and plants. However, microbes can change mechanical pathway and synthesize siderophore which play a crucial role in converting ferric iron oxide to harmless and useful iron [67]. On the other hand, host plants have the ability to strategize and respond to limited iron in the soil by: 

i. Solubilizing the inorganic iron in the rhizosphere appreciably 

ii. Producing plant-siderophores (phytosiderophores) which through transportation are taken to the root tissues via a peculiar uptake system [68]. 

For instance, rice uses the two methods for acquisition of its iron from the soil [68]. An additional strategy by different studies has been proposed for acquiring iron by plants involving the use of iron chelated to siderophores produced by microbes [69]. Studies with fluorescent pseudomonads revealed an ability to produce siderophore which led to the enhancement of iron nutrients in both *Graminaceous* plants and dicotyledonous plant species [70]. Another class of microbiome capable of acquiring iron is the rhizobacteria, e.g., *Bacillus subtilis* GB03, which enhances the ability of plants to acquire their own iron [71]. Fe-deficiency-induced transcription factor1 (FIT1) up-regulation has been reported in another plant called arabidopsis by the same strain of *Bacillus subtilis* by inducing ferric reductase FRO2 and the iron transporter IRT1 up [71]. More details of the microbial rhizosphere mechanism enhancing plants iron uptake have been overviewed [69,72]. Many species of the microbial rhizosphere, especially the rhizobacteria, derive their nutrient carbon, which is the source of energy for them, from organic compounds that have been assimilated in the soil, but these organic compounds are not well degraded and hence carbon is not readily available in the soil and this in turn limits microbial growth. These species of rhizobacteria are referred to as organotrophs [73]. When minerals in the soil are broken down by a group of bacterial microbiome, nutritive cations are released not only for the microbial growth but also to nourish the plant for its growth. Reports have indicated that both rhizosphere and ectomycorrhizosphere are the major abodes for bacteria involved in weathering soil minerals [74] and these have been found to enhance the growth and development of plants even in very low fertility soils [75].

## 8. Plant Growth Promoting Rhizobacteria (PGPR) Effects on Growth and Development of Plants 

For favorable effective contribution towards the growth of a plant and its development, the PGPR style of action generally depends upon the following categories or at least one of them [76].
Bioprotectants: This comprises strains of PGPR acting as biocontrol agents in order to suppress the pathogens and thus prevent plants from diseases or infections [77]. The same mode of action is required by the plant to develop resistance against bacterial [78], fungal [79], viral phytopthogens [80], insects [81] and also nematodes [82]. The ability of PGPR to produce and discharge metabolites which can ameliorate pathogens’ microbial loads and their activities or rhizosphere microflora that are deleterious is another major type of action found in several strains of PGPR [83,84]. For instance, siderophore compounds are produced which cleave ferric iron, causing it to be unavailable or rarely accessible to the inhabiting pathogenic microbes, diffusible antibiotic compounds, volatile organic compounds (VOCs), lytic enzymatic compounds, biosurfactant compounds and toxic compounds [85,86]. Competition for survival of the strains of PGPR with other phytopathogens by competing with the very few nutrients available and the little space in the rhizospheric environment is similarly a common and potential yardstick to checking the unwanted microbes’ growth, inhabiting the rhizospheric environment [87].Biofertilizers: These comprise of the strains of PGPR which enhance the uptake of nutrients by plants thus promoting the germination of seed and seedling development; this leads to crop yield improvement [86,88]. Several and different actions that involved PGPR as biofertilizers are fixation of N_2_ [89], enhancing the availability of phosphorous to plants (through solubilizing the inorganic phosphate and organic phosphate mineralization) [90] and discharging organic acids which aid to make nutrients like zinc and other vital elements available in a usable forms [91]. Biostimulants: PGPR that are capable of producing phytohormone compounds, such as secondary metabolites like cytokinins, auxins, indole acetic acid (IAA), vitamins and riboflavin [91] are described as biostimulants. Moreover, some PGPR have the ability to degrade some complex chemicals like pesticides, herbicides and insecticides which have been found to be phytotoxic. This is a vital trait of the PGPR, for instance, *P. aeruginosa* PS1 ameliorate the toxicity of some herbicides used on leguminous plants such as clodinafop and quizalafop-p-ethyl [92]. In the presence of some insecticides, plant growth enhancing compounds have been reportedly produced and this includes pyriproxyfen and fipronil in some plants [93]. Likewise, reports have shown the *E. asburiae* PS2 strain possessing some vital plant growth enhancing activities such as indole acetic acid, solubilization of phosphate, siderophores production, hydrogen cyanide, exopolysaccharides and ammonium compounds in the presence of some herbicides like glyphosate, quizalafop-p-ethyl, metribuzin and clodinafop [94]. The strain of rhizobial MRL3 is another PGPR that has been shown to possess plant growth promoting traits when the organism was used to treat soil contaminated with insecticides such as pyriproxyfen and fipronil in plant lentil [95]. Hence, it could be concluded that strains of PGPR can be applied for the enhancement of the growth of plants even in soils that have been contaminated for a long period of time with different types of inorganic substances. From these novel traits of PGPR, plant eco-friendly rhizosphere microbes can be classified as either host plant growth enhancing microbes (HPGEM), which have a direct impact on the enhancement of the growth of plant, or as bio-control agents (BCA) which have a significant influence on the health of a plant by inhibiting phytopathogens, and hence have an indirect effect on its growing ability [96]. Thus, PGPR have direct and indirect mechanisms with significant traits that improve plant nutrition and health.


## 9. Bacterial Colonization and Their Systemic Inductive Resistance 

The microbial competition with others to invade and inhabit plant habitats like the rhizosphere is paramount to plant metabolisms [97]. Proliferation of a single colony of bacterial cells is made possible when it attaches to the surfaces, after which cells divide, then it forms dense colonial aggregates of cells which are commonly called macro-colonies or biofilms. The colonization of the bacterial cells includes the following steps: attraction, recognition, adherence and invasion, however, for endophytic and pathogenic organisms it is accompanied with colonization and growth. They devise several strategic means to establish interactions or associations with the plant. Crosstalk is initiated by the plant roots which attract the soil microbes. The microbes are able to recognize the signals and thus the signals initiate the bacterial colonization [86]. PGPR possess an organ called flagella which enhances their motility to the root surfaces and the sense of chemotactic responses guides them to the appropriate parts of the plant root [98]. This suggests that PGPR survival in their habitats would depend either on their capabilities to make use of a specific habitat or their ability to become resilient to changes in the new habitat or plant conditions. For instance, bioinoculation of the strain of S499 of *Bacillus subtilis* to plant seedlings was able to colonize the root system of two different plants effectively and successfully. However, the microbial number of many PGPR inoculants usually reduces progressively in time after inoculation from 10^7^–10^9^ cells per gram of dry soil to 10^5^–10^6^ cells per gram of dry soil after 2–3 weeks [98]. This may be due to the fact that the microbial inoculants need to genetically acclimatize to the new environment. Nevertheless, the microbial load threshold is often enough to bring about beneficial effects on the plants [99]. Biological control microbes will be able to have positive effects by their abilities to effectively colonize, survive and proliferate in the rhizosphere along growing plant roots over a great period of time and in the presence of native micro-inhabitants [100]. 

In the colonization of bacteria, the authors of [101] have revealed the regulatory role of jasmonic acid (JA) and ethylene (ET) phytohormones in the systemic resistance beneficial rhizobacteria often induced in plants. However, some strains of rhizobacteria induce systemic resistance through the salicylic acid (SA) pathway and not via the JA/ET pathway [102]. Other rhizobacteria such as a strain of *Bacillus cereus* AR156 induce systemic resistance via signaling both SA and JA/ET pathways [103]. In addition, several different genes responsible for defense mechanisms in plants like *MPK3*, *MPK6*, *WRKY22*, *WRKY29* and *Pdf1.2* have been shown by [104] to be activated by the molecules from quorum sensing released by the rhizospheric bacteria. Furthermore, a lot of work goes on to understand the mechanism of reaction behind the mystery of a bacterial strain inhabiting a particular rhizosphere conferring a transcriptional and metabolic change on such a host plant. For instance, in the *Arabidopsi* plant, it has been shown that the induction of resistance on the host plant by the rhizobacteria could either be by JA/ET or SA pathway(s). The bacterial strain responsible for induction of resistance using the pathways of JA/ET causes a less noticeable transcriptional transformation on the plant metabolisms [105], while a significant transcriptional change is noticed in plant metabolisms when the same bacterial strain causes an induction of resistance on the host plant [102]. In addition, by combining the understanding of transcriptional profiles and metabolic pathways of immune responses of plants initiated by beneficial bacteria, [106,107] reported different strains of *Pseudomonas fluorescens* that effectively initiated host plant immune resistance against infestation of *P. syringae*; however, they tampered with available carbon in the soil. This literature review shows the crucial role rhizospheric microbiome, especially bacteria, performs in initiating/aiding host plant immune responses against phytopathogenic microbes and the physiological changes in their metabolisms, thus promoting the induction of biosynthesis of some known vital metabolites and unknown novel metabolites that could be of biotechnological importance [102]. Anatomical, structural and other physiological analyses of ‘cryptic’ plant compounds (i.e., systemic resistance) induced by beneficial bacteria known as rhizobacteria should be of great concern for researchers.

## 10. Mycorrhizal Fungi Interaction with Plants

The term mycorrhiza is a Greek word that could be split into myco and rhiza, meaning fungus and root, respectively. Generally, this term describes an association between a fungus inhabiting soil and the root of a plant, which is a symbiotic relationship. In contrast with the rhizobia relationship with their leguminous plants, fungus and root association (mycorrhizal associations) are ubiquitous, cosmopolitan and moderately nonselective and occur in all of the members of gymnosperms and approximately 80% of the members of angiosperms [108]. The ubiquity and cosmopolitanism of mycorrhizae throughout the kingdom of plants could be traced to the capability of plants to reject fungi, which has been in existence for approximately 450 million years. Majorly, these associations are usually beneficial to both the hosting plants and fungal colonizers; however, neutral and parasitic associations do exist. Mycorrhizal association is very crucial in the acquisition of nutrients for plants; for instance, plants are able to obtain water and important nutrients like phosphorus via the association. It also assists in acquiring micronutrients like Cu and Zn and others form the organic compounds in the soil and consequently carbon is released from the plants for the sustenance of the association. Physical interface with the host plants is used to broadly categorize mycorrhizal association into endomycorrhiza and ectomycorrhiza. Primarily, ectomycorrhizal fungi are present in the plant roots that are woody such as the trees in the forest, they form a hypha-densed covering called a fungal mantel/sheath which is above the tip of the root. It is from this that the hyphae develop into the intercellular spaces, which now form a net that is referred to as hyphae Hartig net around the cortex cells of the root, though it does not penetrate into the wall of cells. In contrast, the hyphae of endomycorrhizal fungi grow into the cortex of the root and penetrate the cells and this forms a structure known as arbuscule. Arbuscule is a highly branched fan-like structure that plasma membranes of a plant separate from the cytoplasm [109]. Additionally, endomycorrhizal fungi could also be categorized widely into arbuscular mycorrhiza (AM), specialized ericoid and orchid mycorrhiza and according to their names they are colonizers of the species of these particular plants, for example cranberry. Among all the associations of mycorrhiza, the AM fungal relationship is the most abundant. In the arbuscules and Hartig net cases, there are nutrients and carbon transfers to the plant and fungus, respectively, because the contact area between the plant and fungus is increased by the two associations. Furthermore, during mycorrhizal associations both endomycorrhiza and ectomycorrhiza demand about 20–40% of the total fixed carbon plants synthesized during photosynthesis; however, ectomycorrhiza do not absolutely depend on the plant while the endomycorrhiza is totally dependent on the plant.

In the mycorrhizal association, the chemical signals that usually initiate the relationship with plants are complex and yet to be understood, like the Rhizobial associations. It could be said that, because endomycorrhiza form an obligate symbiotic relationship with their host plants, that is why the association is yet to be understood. This means that they could not be cultured independently of their hosts; hence it is relatively impossible for researchers to study them under controlled parameters in the laboratory. Interestingly, ectomycorrhizal symbiotic relationship development is more researched and understood because the organisms involved could be cultured independently. Obviously, in the two cases, the fungus senses the plant root exudates where volatile chemical carbondioxide is present, which the branching and hyphal growth usually initiate [110]. On the other hand, chemical exudate perceived by the plants from mycorrhizae have been supposed to exist for a long time, but recently one of the chemical structures of the ecotmycorrhizal association have been completely characterized [111,112]. Furthermore, these chemical signals are usually referred to as *Myc*-factors and are structurally similar to the nod factors that are aptly generated by rhizobial microbes during the symbiotic association between rhizobia and leguminous plants. The production of fungal signals is still not understood if it is produced only when it senses that the host plant is close or the signals are present all the time. Nevertheless, the spores produced by the fungi in the soil are dormant until the germination and growth of hyphae are activated by one of the stimuli. The growth of hyphae and branching are increased if the fungi are nearby the root and presumably branching of the root is initiated by *Myc*-factors to enhance the chances of interception of the root fungus. 

In addition, contact with the root of a plant causes infection, which could be in arbuscular mycorrhizae with the formation of a hyphopodium or ectomycorrhizea, which is the growth between dermal cells. The fungal hyphae cease develops if the root is not in the surrounding area; this makes the sporulated fungus go back to the vegetative stage and thus its triacylglyceride and glycogen reserves are maintained. The interception of a root is achieved via multiple instances of initiation of the growth of hyphae. In order for nutrients to be acquired for the plants, the hyphae length of both endo- and ectomycorrhizal fungi project into the surrounding soil which lead to a tenfold increase in the root surface area of a plant that is effective. Moreover, per unit of length, it results in a two to threefold increase in the phosphorus uptake and other nutrients, compared to other plants that are not mycorrhizal. Moreover, the acquisition of nutrients is not solely dependent on the quantity of hyphae but also their tiny size, which is less than 200 mm; this assists in accessing the pores and cracks in the soil, which are ordinarily impenetrable for the plant root. It is worth noting that the soil quality is largely dependent on the hyphae of the fungi network, which emanate from the roots of the plant. The hyphae of mycorrhizae use several mechanisms like biochemical, physical and biological means to enhance the stability and aggregation of the soil, and this ameliorates soil erosion, increases water percolation and soil aeration and thus the productivity of the plants is improved [113]. The term ‘common mycorrhizal network’ (CMN) refers to an intertwined and dense network of fungal hyphae in the plant root, and these mycorrhizal hyphae are interconnected in two or more plants. It has been shown that plants interact with CMN for the purpose of acquiring nutrients [114]. The kind of mycorrhizal fungi involved in the interaction and how diverse the interaction is have been discovered to be crucial and paramount for the stability and function of the ecosystem and biodiversity of plants [115]. In addition, it has been demonstrated that some classes of fungi have the capacity to make iron available for the plant, for example, *Rhizopus arrhizus*, a fungal siderophore production called rhizoferin which can effectively carry iron to plants with efficiency comparable to that of the artificially synthesized chelates [115].

## 11. General Significance of Archaean Microbe

Archaea domain is another important domain that could serve as an alternative means of enhancing agricultural productions because of their unique characteristics. The archaea groups of microbes, which are cosmopolitan, have been found inhabiting every environment, including extremophilic areas like the arid and semi-arid regions. These groups of microbes are found in different habitats and they are significant in the nutrient (nitrogen, sulfur and carbon) recycling that is very important in agricultural productions because these nutrients are required by the plant in large quantities. In the nitrogen cycle, archaeal microbes have been established to take part in several metabolic steps. One of the reactions is the one that leads to nitrogen removal from an ecological niche, for example, denitrification, respiration based on nitrate and the other reactions introduce nitrogen in the ecosystem, for example, nitrogen fixation and assimilation of nitrogen [116,117]. Recently, studies have revealed the immense importance of archaeal microbes in the reactions of ammonia oxidation, particularly in the oceans and soil environments [118,119]. In the metabolic chains, nitrite is produced by archaea; this is then oxidized to nitrate by other classes of microbes and is then utilized by both microbial consumers and plants [120]. 

Sulfur is another nutrient required by plants in large quantities and has been shown to be recycled via archaeal microbes. In the recycling of sulfur, the element is made available to other microbes when the archaeal microbes that grow in the environment oxidize sulfur compounds and are released into the environment. Methanogenic archaea in the carbon cycle contribute immensely to the decomposition of organic matter by removing hydrogen. In an ecosystem that is without oxygen, these archaeal microbes perform as decomposers, for instance, sewage-treatment, marshes and sediments are broken by these microbes in the ecological zones. The association between archaean communities and a range of other organisms has been proven; for instance, they are found in the zones of rhizospheric plant roots and on the surface of coral [121,122,123]. Archaea also serve as hosts for a novel class of antibiotics that are potentially useful. Within the *Sulfolobus* and *Haloarchaea* classes of archaea, a number of archaeocins have been identified and characterized; however, it is believed that hundred more still exist. Structurally, these compounds are different from that of the antibiotics produced by bacteria and hence their mode of action may be novel. Furthermore, in the molecular biology of archaea, new selectable markers may be created for use [124]. 

## 12. Conclusions

Despite the fact that the vitality of the microbiome inhabiting the rhizosphere and their effective roles in the plant ecological environment have been greatly and broadly acknowledged, there are limitations to the conventional techniques in their capacity to unravel the diversity and functions of these microbiome in their ecosystem and also no knowledge exists for the enormous majority of rhizospheric microbiome. In order to really explore and have better insights into the diversity, complexity, community structure, ecology and physiology of rhizospheric microbiome, conventional techniques have to be coupled with metagenomics, which is a new and advanced technique of next generation sequencing. Chemical and microbial markers to illuminate whether and how plants absorb and facilitate beneficial microbes are provided when the chemical compositions in the plant rhizosphere, like the exudates, key players and signals, are identified. A better understanding of the rudimentary principles of rhizosphere ecology, including the diversity and functions of inhabiting microbes, is on the way, especially in the use of metagenomics and various bioinformatics tools. However, further knowledge is indispensable for the optimization of soil microbial technology to benefit the growth, development and health in the usual environment. In general, this can constitute overwhelming substantiation indicating that a persistent manipulation and exploitation of plant growth promoting rhizobacteria (PGPR), mycorrhizal fungi and archaeal microbes could be a great future breakthrough in agricultural sustainability. Consequently, current approaches to agricultural productions such as the indiscriminate application of chemical fertilizers and pesticides with a lot of short and long term environmental and health challenges will be ameliorated. Therefore, research should focus on the culture independent method, that is to say metagenomics and bioinformatics, which have the ability to discover the entire diversity and all the functions and genes of microbes (especially archaea) that could be of biotechnological benefits, should be embraced for the purpose of improving agricultural productions. 

## Figures and Tables

**Figure 1 ijerph-16-03873-f001:**
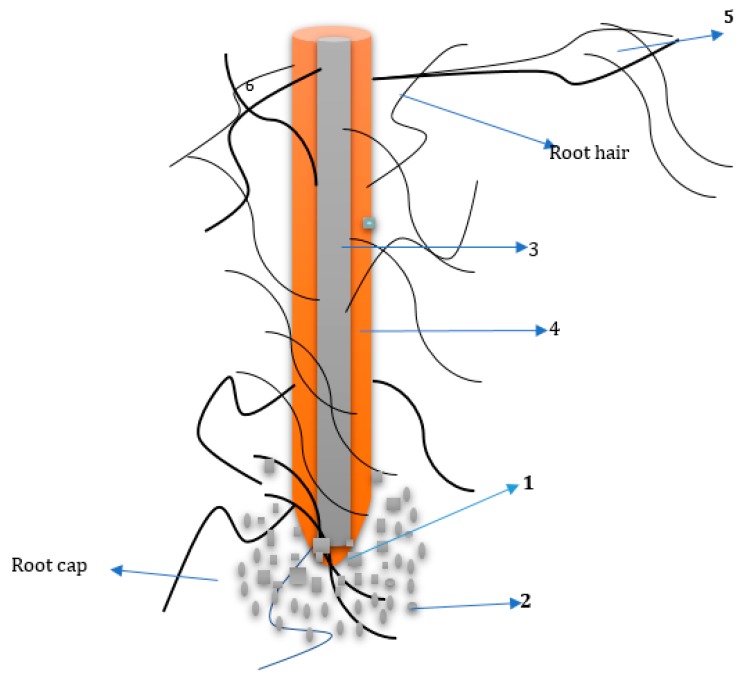
A typical diagram of a plant root representing the six main zones of rhizodeposits. (1) Lysis of lid and marginal cells, (2) lysis of complex and impenetrable mucilage, (3) lysis of simple and penetrable root exudates, (4) lysis of volatile organic compounds, (5) lysis of carbon to mutualists and (6) lysis of carbon due to root epidermal and cortical cell death.

**Figure 2 ijerph-16-03873-f002:**
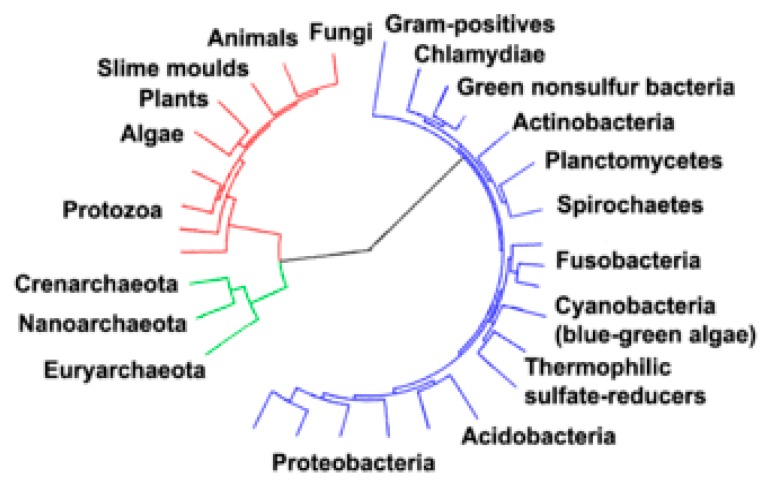
Phylogenetic tree showing diversity for each of the three domains of life. The bacterial domain is blue, eukaryal domain is red, while the archaeal domain is green [56].

**Table 1 ijerph-16-03873-t001:** Various complex exudates released via the plant root and their single constituent.

Complex Exudates	Compound Constituents
Organic compounds	Succinic acid, *l*-aspartic acid, Acetic acid, *l*-glutamic acid, salicylic acid, malic acid, isocitric acid, chorismic acid, shikimic acid, sinapic acid, shikimic acid, *p*-hydroxybenzoic acid, gallic acid, caffeic acid, protocatacheuic acid, *p*-coumaric acid, tartaric acid, ferulic acid, oxalic acid, citric acid, piscidic acid, mugineic acid
Complex carbohydrate	Glucose, arabinose, galactose, sucrose, fructose, pentose, raffinose, rhamnose, ribose, xylose and mannitol
Amino acids	Complete 20 protein genic amino acids, *l*-hydroxyproline, mugineic acid, amino butyric acid, homoserine
Coumarins	Umbelliferone
Flavonols	Kaempferol, quercitin, naringenin, naringin, rutin, myricetin, strigolactone, genistein and their derivative sugars
Lignins	Benzoic acid, nicotinic acid, catechol, cinnamic acid, gallic acid, phloroglucinol, syringic acid, sinapoyl aldehyde, ferulic acid, coumaric acid, vanillin, chlorogenic acid, quinic acid, pyroglutamic acid, sinapyl alcohol
Anthocyanins	Delphinidin, pelargonidin, cyanidin and their derivatives sugar molecules
Aurones	Sinapoyl choline, benzyl aurones synapates
Glucosinolates	Desuphoguconapin, desulphoprogoitrin, cyclobrassinone, desulphoglucoalyssin, desulphonapoleiferin
Sterols	Sitosterol, stigmasterol, campestrol
Anthocyanins	Delphinidin, pelargonidin, cyanidin and their derivative sugar molecules
Fatty acids	Oleic acid, linoleic acid, stearic acid, palmitic acid
Indole compounds	Brassitin, sinalexin, indole-3-acetic acid, methyl indole carboxylate, camalexin glucoside, brassilexin
Proteins and enzymes	Lectins, proteases, PR proteins, peroxidases, phosphatases, lipase, hydrolases
Allomones	Sorgoleone, 5,7,4′-trihydroxy-3′, jugulone, DIMBOA, 5′-dimethoxyflavone, DIBOA

**Table 2 ijerph-16-03873-t002:** Microbial Domains Comparisons [48].

Property	Bacteria	Archaea	Fungi
**Cell Membrane**	Made up of peptidoglycan and lipids are linked via ester molecule,	Made up of pseudo-peptidoglycan and lipids are linked via ether molecule	Made up of different structures and lipids are linked via ester molecule
**Gene Structure and Configuration**	Chromosomes are circular, translation and transcription are unique	Chromosomes are circular, translation and transcription are similar to eukaryotes (fungi)	Chromosomes are multiple and linear, translation and transcription are similar to archaea
**Structure of Internal Cell**	The nucleus or organelles has no membrane bound	The nucleus or organelles has no membrane bound	There is membrane bound nucleus and organelles
**Metabolic Reaction**	There are several, including aerobic and anaerobic respiration, photosynthetic, autotrophic reactions and fermentation	There are several with methanogenic reaction specifically unique to this domain	Cellular respiration, fermentation and photosynthetic reaction
**Reproduction**	Reproduction is asexual and transfer of genes is horizontal	Reproduction is asexual and transfer of genes is horizontal	Reproduction is sexual and asexual

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
