# Peer review of "Bacteria, Fungi and Archaea Domains in Rhizospheric Soil and Their Effects in Enhancing Agricultural Productivity"

_ijerph, 2019, doi:10.3390/ijerph16203873_

Round 1

Reviewer 1 Report

This revision is quite complete and it will be useful for authors working in this field and also for researchers just beginning in this topic.

My major comment is concerning the introduction of a new term, HPGER (host plant growth enhancing rhizobacteria), that is basically the same that PGPR (plant growth promoting rhizobacteria). I do not see the need of introducing a new term when PGPR is widely used and accepted by the scientific community. It could be confusing. Even more, the use of “host plant” suggests certain specificity in the plant-rhizobacteria interaction and most of the time there is no specificity and the same bacteria are found in the rhizosphere of even distant plants. This is just an opinion, but I think that we should not introduce confusion in the audience.

Although English is not my mother tongue and the work, in general, is readable, I feel some sentences are not clear enough so I suggest a revision by and English speaking person.

Author Response

I have made the corrections of changing HPGER back to PGPR in the manuscript.

Reviewer 2 Report

Similar review and original research articles are easily found, e.g., The rhizosphere microbiome: significance of plant beneficial, plant pathogenic, and human pathogenic microorganisms. Mendes R et al. FEMS Microbiol Rev. (2013); An Ecological Loop: Host Microbiomes across Multitrophic Interactions. Liu H, Macdonald CA, Cook J, Anderson IC, Singh BK. Trends Ecol Evol. 2019 Aug 15; Role of Plant Growth Promoting Rhizobacteria in Agricultural Sustainability-A Review. Vejan P et al. Molecules. (2016), etc.

The second part Rhizosphere is too long and redundant. Introducing too much basic knowledge is not necessary. There is a lack of vivid graphics. In “The Effects of Rhizosphere”, only one paragraph is presented. There is a lack of comprehensive and in-depth summary of this field of recent research. The summary in Table 1 is disappointing; there are only two columns, and the references cited are very old, published in 2000 and 2003 respectively. Actually most references are not up to date.

Table 2 and Fig. 1 might be suitable for use in textbooks rather than in a review paper.

“11. Effects of Beneficial Bacteria in Bioremediation” is too brief and has no reference value. Such blindly written parts should be deleted.

This review tries to cover too many aspects, and no deeper insights are discussed in many parts, e.g., 12. The colonization of Bacteria with Plant, and 13. Mycorrhizal Fungi Interaction with Plants, etc.

In “14. General Significance of Archaean Microbe”, the summary and review are shallow and not relevant. What are the far reaching effects of Archaea on the agricultural productivity?

In “15. Conclusion”, no prospects and innovative perspectives toward future work are presented. What are your own experiment results, idea and suggestions of this field?

There are numerous language errors, e.g., “......dissolved oxygen can be bioassay by the yield of the crop......”, “......carbon in the surroundings plays a great a role in determining......”, etc.
